# Big Five Personality Model-based study of death coping self-efficacy in clinical nurses: A cross-sectional survey

Xi Lin [1,2]☯, Xiaoqing Li[1]☯, Qing Liu[1]*, Shengwen Shao[2], Weilan Xiang[3]

**1** Department of Pediatrics, The Affiliated Hospital of Southwest Medical University, Luzhou, Sichuan, China, **2** Department of Nursing, Huzhou University, Huzhou, Zhejiang, China, **3** Department of Nursing, Zhejiang University School of Medicine Sir Run Run Shaw Hospital, Hangzhou, Zhejiang, China

☯ These authors contributed equally to this work.
* 1130610822@qq.com

## Abstract

### Background

Specific personality traits may affect the ability of nurses to deal with patient death. The relationship between personality and death coping self-efficacy (DCS) has rarely been investigated in the palliative care setting. In this study, we explored the associations between different personality profiles and DCS in clinical nurses from general wards and ICU.

### Methods

A cross-sectional survey of 572 Chinese nurses was conducted between August and September 2020, by way of a self-administered questionnaire.

### Results

Among the Big Five Personality Traits, in nurses the score was highest for conscientiousness and lowest for neuroticism. With regard to DCS, nurses scored highly on the intention of hospice care. The Big Five Personality Traits were found to explain 20.2% of the overall variation in DCS. Openness, agreeableness and conscientiousness were significantly associated with DCS in nurses.

### Conclusions

Nursing managers should pay attention to differences in personality characteristics and provide personalized and targeted nursing education. This should improve nurses' DCS, enrich their professional development and promote high quality palliative care for patients and their families.

**Data Availability Statement:** All relevant data are within the paper and its Supporting information files.

**Funding:** The author(s) received no specific funding for this work.

**Competing interests:** The authors have declared that no competing interests exist.

## Introduction

The demand for palliative care in China is expected to increase significantly due to the aging population. Achieving a "good death" has traditionally been the core goal of palliative care, which aims to optimise the individual's dignity and quality of life [1, 2]. However, according to a 2015 Economist Intelligence Unit survey, only 0.3% of the population in mainland China has access to palliative care prior to death, ranking China 64th out of 80 countries in terms of its capacity to provide this service [3]. Chinese attitudes toward death are significantly different to those of Western countries and are influenced by traditional culture. This extends to the doctors and nurses engaged in healthcare provision, who may avoid broaching the topic of death [4]. Moreover, patients and medical institution administrators generally believe that dealing with death-related subjects is part of the mission of health care. Consequently, the psychological stress and trauma experienced by medical staff and by front-line nurses in particular is often ignored [4]. Frequent exposure to death-related events nevertheless leads to compassion fatigue and burnout [5]. Such events may therefore adversely affect the physical and psychological well-being of health care workers [6]. This in turn can compromise productivity, performance, staff turnover, and even quality of care [7].

Bandura (1977) proposed a social cognitive theory in which self-efficacy, i.e. the belief in one's ability to exercise control over one's own functioning and over events that affect one's life, is the best predictor of behavior [8]. Personal self-efficacy also affects the expectation of results Bugen (1980) then classified death coping self-efficacy (DCS) into two categories: the ability to cope with oneself and the ability to cope with others. The former includes the ability to understand death, master the knowledge of death, and express emotions related to death, while the latter encompasses the ability to communicate with the dying and bereaved [9]. DCS was defined by Robbins (1992) as a set of abilities and skills that people use to deal with their own death and with the death of others, as well as their attitudes and beliefs towards these abilities [10]. In other words, DCS refers to the positive adaptive response that individuals use to reduce stress when coping with the event of death. Robbins defined four dimensions of DCS: the ability to cope with death and loss, understanding grief and support systems, death preparation behavior, and the willingness to talk about and to face death.

In China, nurses are the evaluators, educators and implementers of palliative care and thus play an important role in the comprehensive care of end-stage patients and their families [11]. As the health care professional having the most contact with terminally ill patients, nurses are the primary facilitator of patient data assessments. The results provide a reference basis for nursing staff to formulate targeted nursing intervention measures. They also form the evaluation basis for clinicians to formulate and modify the patient diagnosis and treatment plan [11]. Nursing has expanded in depth and breadth since its' beginning as a medical auxiliary discipline into an independent discipline. The palliative care specialty has a strong practical basis and its clinical practice is challenging due to the particularity of its patients. Nursing staff are constantly learning by experience during practice. As educators they pass on this knowledge and experience in two ways [12]. Firstly, nurses are the primary implementers of health education during the process of patient care. Secondly, nurses are also educators for the nursing profession.

Due to resource limitations, there is currently a shortage of psychologists, social care workers and volunteers within multidisciplinary teams in hospitals, with very few nurses professionally trained to provide emotional support [12]. A survey of 1,680 nurses and nursing assistants on the topic of end-of-life communication found a low level of self-efficacy when nurses discussed the course of disease or prognosis [13]. This indicates that nurses experienced difficulty during end-of-life communication and lacked the relevant communication skills.

Various emotional experiences including fearfulness, guilt and self-blame emerged in nurses following patient death [14], suggesting they lacked the ability to cope with death. Nurses who cannot cope with patient death may be unable to support dying patients and their families, thereby reducing the quality of palliative care. Consequently, low DCS in nursing is not compatible with good palliative care. Recent studies have found that some of the main barriers to palliative care include angry family members, poor understanding of nursing care by family members, sensitive physician behavior, and the relationships between nurses [13, 15]. In preparation for patient death, nurses are expected to show active concern for the imminent death facing the patient, to separate their emotions from their work, and to address any conflicts that occur [16].

Palliative care knowledge and nursing experience are key to achieving palliative care self-efficacy [17]. In a study by Pfister *et al.* (2013), knowledge of palliative care was positively correlated with work experience, whereas self-efficacy was negatively correlated with age and work experience [18]. More recently, Evenblij *et al.* (2019) found that nurses' end-of-life care was positively associated with work experience, older age (>36 years), formal training in end-of-life care, and knowledge [19]. In summary, nursing staff self-efficacy in the setting of palliative care may be influenced by their age, work experience, work unit, willingness and attitude to care for patients, and whether they have formal palliative care training and related knowledge. However, some divergent findings remain.

Over the past half century, the Big Five Personality Model (neuroticism, conscientiousness, openness, agreeableness and extraversion) has been extensively studied and proven to be stable across languages, cultures, and assessors. It has therefore been widely accepted by personality psychologists at the dimensional level [20]. Personality characteristics are a unique integrative mode of thought, emotion and behavior displayed by individuals. These characteristics can have an impact on individual job type, job burnout degree, and job engagement [21]. The sense of responsibility and the activity of job engagement in the nurses' Big Five Personality Model are significantly correlated with high conscientiousness [22–24], and with low neuroticism, high extraversion, and positive prediction of well-being [24]. It has also been reported that nurses with higher neuroticism and lower agreeableness have an increased risk of depression and anxiety [22, 25]. Nurses with death-related anxiety and depression are less competent in death-related work. It has also been shown that nurses with high extroversion appear to have higher general self-efficacy when facing stress compared to others [26].

Few studies have examined the relationship between personality traits and coping strategies in the context of death events. Stevens [27] conducted a study of 40 college students who had experienced the death of a significant other in their lives with the aim of exploring the relationship between life experience and death coping strategies. They found that emotion-focused coping strategies were positively correlated with low life meaning. Robinson *et al.* [28] studied 138 bereaved mothers and evaluated the influence of personality factors and coping factors on grief intensity. They found that a high level of neuroticism or the use of more task-oriented coping strategies could affect a mothers' grief following the death of their child. A Taiwanese study by Hsu [29] of 556 university nursing students showed that high levels of openness, agreeableness and conscientiousness, as well as low neuroticism, were associated with the ability to talk about death, the ability to process loss, the intention to increase meaning in life, and the ability to process funeral practices. This suggests that high levels of neuroticism and task-oriented coping strategies are less effective in dealing with significant loss or the death of a significant other. In conclusion, there is evidence in the literature for a correlation between coping behaviors and personality traits. Different personality traits may adopt different coping strategies in the face of death.

While several studies have explored the impact of personality traits on coping and self-effi-
cacy [30–32], to our knowledge there are no studies on the relationship between personality
traits and DCS. Although personality traits can affect people's behavioral responses, it is not
known whether self-efficacy in the circumstances surrounding the event of death is also
affected by personality traits. Also unknown is whether personality traits play a role in explain-
ing DCS in nursing. Therefore, based on the existing literature we hypothesized that different
personality traits could be correlated with DCS. In the present study we explored the relation-
ship between personality traits and DCS in nurses from general wards and ICU.

## Methods

### Study design and participants

A cross-sectional survey design was used. Convenience sampling was used to recruit partici-
pants from a tertiary hospital. The inclusion criteria were: (1) nurses working in general wards
and ICUs; (2) registered nurses having worked in a nursing position for at least one year; (3)
able to understand the purpose of the study and voluntarily provide written, informed consent;
and (4) be directly involved in patient care.

A total of 2450 nurses were invited to participate, of whom 755 responded to the survey. Of
these, 572 responses were valid after data cleansing, giving a response rate of 23.3%. Of the 183
invalid and excluded cases, 137 responded with obviously biased or poorly considered
responses (e.g., providing the same response to all the questions for at least one scale that
included reverse-score items), and 46 did not provide a response for at least two-thirds of the
questionnaire.

### Measurements

**General characteristics.**    We assessed the general characteristics of the nurse participants
using a questionnaire that asked about age, gender, marital status, educational background,
religious beliefs, number of years of experience, personal bereavement experience, whether
they had received death-related education, and their attitude to talking about death.

**Death Coping Self-Efficacy Scale.**    The ability of nursing personnel to cope with death
was assessed using the Death Coping Self-Efficacy Scale (DCSS), as adapted from the Hospice-
Related DCSS by Robbins [10]. This was previously translated into Chinese with good reliabil-
ity and validity in a Taiwanese sample [33]. The DCSS consists of three dimensions with a
total of 29 items rated on a Likert scale from 1 to 5 points (1 = highly uncertain, 5 = completely
certain). These items comprised hospice care (12 items), coping with grief (9 items), and prep-
aration for death (8 items). The response level of nurses to death was positively correlated with
their scores. Higher scores indicated the adoption of more effective adaptive strategies by the
nurses. Authorization from the original developer of the DSCC scale was obtained for this
study. Guillemin's [34] basic criteria for scale cultural adjustment and cross-cultural adjust-
ment guidelines were strictly followed in the translation. The complex Chinese version was
revised to create a simplified Chinese version following consultation with 5 nursing experts
from the hospital and school. Thirty clinical nurses were then pre-surveyed to determine the
conceptual equivalence and content equivalence of the scale.

The item-level content validity index of the simplified Chinese version of the DSCC ranged
from 0.80 to 1.00, the scale-level content validity index/average agreement was 0.97, and Cron-
bach's α for scale was 0.903, 0.909, 0.860 and 0.820 for the subscales.

**Chinese Big Five Personality Inventory Brief Version.**    Big Five personality domains
were assessed using the Chinese Big Five Personality Inventory Brief Version (CBF-PI-B). This
is a well-validated, 40-item self-report instrument for the Chinese population that describes

five personality dimensions: neuroticism, extraversion, openness, agreeableness and conscientiousness [35]. The CBF-PI-B includes 8 items within each dimension, rated on a 6-point Likert scale (1 = totally disagree, 6 = totally agree). Higher scores indicate higher personality trends for the corresponding dimensions. Scores for the 8 items are reversed before being summed for each personality dimension. The content validity index was 0.92. The Cronbach's α for the Chinese version used in this study was 0.74~0.89, hence these scales exhibit good reliability and validity.

## Data collection and ethical considerations

Prior to data collection, ethical approval was obtained from the Research Ethics Committee of the Sir Run Shaw Hospital, College of Medicine, Zhejiang University (No.20201029-31). Data were collected from August 10 to September 14, 2020. With the cooperation of unit managers and nursing units, researchers visited the wards and instructed subjects on how data would be collected. Using a convenient sampling method, envelopes containing a questionnaire and a declaration of written consent were provided to each participant. The aim of the study was explained and the voluntary and confidential nature of this research was emphasized. Two weeks after the distribution of envelopes, the researchers revisited the wards and collected the completed questionnaires. At the end of the study, nurses received a gift voucher by way of thanks for their participation.

## Data analysis

Prior to analysis, the data was checked on the assumptions of normality and multicollinearity. Data were recorded and analyzed using SPSS version 26.0 and were described using mean and standard deviations (SD) for continuous variables and percentages for categorical data. One-way ANOVA tests were used to evaluate differences in personality profiles among the DCS. Pearson's correlations were used to examine relationships between personality profiles and DCS. Finally, multiple regression analysis was used to regress these difference scores against personality scores.

## Results

### Participant characteristics

Of the 572 nurses who successfully completed the survey, 569 (99.5%) were women and more than half (57%) were married. Their mean age was 32.4 ± 7.1 years (range, 23–56) and mean clinical experience was 8.10 years (SD, 6.42; range, 0.1–36). Forty-three participants (7.5%) had a Master's degree and most (94.8%) did not have a religious affiliation. Detailed socio-demographic characteristics for the study cohort are shown in Table 1.

### Death Coping Self-Efficacy and Big Five Personality scores

Table 2 shows DCSS scores for each dimension, as well as the total DCSS score. Total mean scores on the Likert scale are given, as well as mean item scores reflecting the response to individual items.

Table 3 shows mean DCSS scores according to the predominant Big Five Personality Traits. The differences in mean scores between the Big Five Personality Traits were statistically significant ($p$ = .02). The highest DCSS score was seen with agreeableness and the lowest with neuroticism.

**Table 1. Characteristics of participants (N = 572).**

| Variable | Category | Frequency (N) | Percentage (%) |
|---|---|---|---|
| Gender | Male | 3 | 0.5 |
| | Female | 569 | 99.5 |
| Marital status | Single | 238 | 41.6 |
| | Married | 326 | 57.0 |
| | Divorced | 8 | 1.2 |
| Age | ≤30 | 348 | 60.8 |
| | 31~40 | 187 | 32.7 |
| | ≥41 | 37 | 6.5 |
| Length of service (yrs) | ≤10 | 418 | 73.1 |
| | >10 | 154 | 26.9 |
| Department | Surgical | 213 | 37.2 |
| | Medicine | 245 | 42.8 |
| | ICU | 28 | 4.9 |
| | Oncology ward | 52 | 9.2 |
| | Emergency | 34 | 5.9 |
| Educational background | Associate (College) | 11 | 1.9 |
| | Bachelors (University) | 518 | 90.6 |
| | Masters | 43 | 7.5 |
| Religious affiliation | Yes | 30 | 5.2 |
| | None | 542 | 94.8 |
| Received death-related education | Yes | 352 | 61.5 |
| | No | 220 | 38.5 |
| Personal bereavement experience | Yes | 202 | 35.3 |
| | No | 370 | 64.7 |
| Attitude in talking about death | Feeling uncomfortable | 158 | 27.6 |
| | Trying to avoid | 74 | 12.9 |
| | Quite open | 34 | 59.4 |

Note: DCSS, Death Coping Self-Efficacy Scale; ICU, intensive care unit

## Correlations between different personality traits and death coping self-efficacy

Table 4 shows correlations between the Big Five Personality Traits and different dimensions as well as with the total DCSS score. Extraversion, openness, agreeableness and conscientiousness were all positively associated with hospice care, coping with grief, preparation for death and with the total DCSS score. On the other hand, neuroticism was negatively associated with hospice care, coping with grief and the DCSS total score.

**Table 2. Death coping self-efficacy mean scores of nurses (n = 572).**

| Variable | Total mean score | Mean item score |
|---|---|---|
| Hospice care | 47.43±5.81 | 3.95 ±0.48 |
| Coping with grief | 27.14±4.50 | 3.39± 0.56 |
| Preparation for death | 28.01±5.58 | 3.11±0.62 |
| Total DCSS score | 102.58±12.07 | 3.48±0.42 |

**Table 3. Death coping self-efficacy scores of nurses with different personality traits (n = 572).**

| Variable | The DCSS score Mean (SD) | Frequency (N) | F | P |
|---|---|---|---|---|
| Neuroticism | 96.80±12.48 | 8 | 2.76 | 0.02 |
| Extraversion | 98.62±7.22 | 36 | | |
| Openness | 102.58±12.07 | 131 | | |
| Agreeable | 103.97±12.28 | 161 | | |
| Conscientiousness | 102.72±11.95 | 236 | | |

**Table 4. Correlations between the Big Five Personality Traits and death coping self-efficacy scores.**

| Variables | Neuroticism | Extraversion | Openness | Agreeable | Conscientiousness |
|---|---|---|---|---|---|
| Hospice care | -.185** | .313** | .466** | .346** | .407** |
| Coping with grief | -.089* | .178** | .019** | .207** | .204** |
| Preparation for death | .051 | .150** | .105* | .267** | .196** |
| Total DCSS score | -.111** | .289** | .314** | .362** | .363** |

Note:

*$p < .05$

**$p < .001$

## Regression analysis of death coping self-efficacy

The Big Five Personality Traits accounted for 20.2% of the overall variation in DCS. Agreeableness, openness and conscientiousness were significantly associated with DCS (Table 5).

## Discussion

This study explored associations between DCS and the Big Five Personality Traits among Chinese nurses working in general wards and ICU. The finding of a lower frequency of neuroticism and higher frequency of conscientiousness in the personality traits of nurses is mostly consistent with previous research [36].

Analysis of the data revealed that high scores for openness, friendliness, extraversion and conscientiousness in nurses' personality traits correlated with higher scores for hospice care, coping with grief and preparation for death. In contrast, neuroticism was negatively correlated with DCS. These results are supported by previous research [37, 38] which found that anxious individuals with high levels of neuroticism are more likely to adopt emotional focus coping

**Table 5. Regression analysis of death coping self-efficacy in nurses.**

| Independent variables | B | SE | β | t (p) | $R^2$(adj $R^2$) | F(p) |
|---|---|---|---|---|---|---|
| Constant | 64.099 | 4.311 | | 14.870(< .001) | 0.209(0.202) | 29.831(< .001) |
| Neuroticism | -.735 | .535 | -.055 | -1.373(.170) | | |
| Conscientiousness | 3.091 | .854 | .173 | 3.618(< .001) | | |
| Openness | 2.214 | .874 | 2.534 | 2.534(0.012) | | |
| Agreeable | 3.682 | .754 | .229 | 4.860(< .001) | | |
| Extraversion | 1.235 | .834 | .068 | 1.481(.139) | | |

Note: β: beta values; SE, Standard error (unstandardized regression coefficient).

Tolerance: .610–.862

strategies and other negative coping strategies. They are less stable emotionally and have personality characteristics such as anxiety, depression, hostility and impulsivity, while also having poor problem solving and self-regulation abilities [39]. Emotional focus coping strategies are statistically correlated with lower life meaning, job satisfaction and identity [40]. In addition, many empirical studies have shown that conscientiousness is positively correlated with good coping behaviors, while neuroticism is positively correlated with stress [41]. Consequently, nursing managers may reduce the stress experienced by nurses with higher neurotic traits by providing them with more social support.

This study found that conscientiousness (41.3%) and agreeableness (28.1%) were the most common personality traits of nurses. Conscientiousness reflects a higher degree of self-control and self-discipline, and such individuals are more careful, attentive, organized, planning-oriented and determined [42]. Studies have found that people with high conscientiousness tend to use active coping, planning, positive reinterpretation and problem coping, while using less denial, psychological and behavioral escape, alcohol and drugs [31]. Conscientious nurses have less frequent missed care [43] and lower job burnout [25], resulting in a higher quality of professional life [23]. This may also result from their tendency towards risk avoidance and self-regulation [43].

Rapport-building is an important characteristic in nurses with respect to the care of patients nearing death and for cooperation with these patients and their families [31]. Nurses with higher levels of agreeableness are more likely to be willing to help and trust others, as well as being more empathetic, kind and polite to patients. Moreover, they are better able to empathize and communicate with dying patients [44], be willing to spend time with them, pay attention to their inner feelings [45], and help them to think about death so as to reduce anxiety in the face of death [46]. A previous study showed that nurses with higher levels of agreeableness also score more highly for job performance, job satisfaction and core competency [47].

Openness as a personality trait refers to openness to experience [48]. It includes tolerance and exploration of strange things, a more active imagination, a willingness to think about new ideas, divergent thinking, and curiosity [49]. General self-efficacy is closely related to the characteristics of post-traumatic growth and can positively stimulate it, while extrovertedness positively predicts the level of posttraumatic growth [50]. Openness to new experiences enables individuals to effectively manage the uncertainty of life, recognize the sustainability of change, and to develop during the process of change instead of being submissive [51]. The results of the present study indicate that nurses with high openness use a variety of methods to solve nursing problems when facing death situations, communicate with patients with optimistic emotions and rich emotional experiences, and improve the medical experience of dying patients.

Our findings provide additional knowledge on the relationship between Big Five Personality Traits and DCS, with personality traits independently explaining 20.2% of the identified differences in DCS. However, the weak correlation between these two parameters, as evidenced by the low value of the Pearson's correlation coefficient, suggests that the interpretation of these results has limitations. While the relationship between personality traits and DCS was statistically significant, the explanatory power is weak and thus any inference would be inappropriate. It can be concluded that the personality traits of clinical nurses is not the only factor that determines their death coping ability. Although personality is an indicator of the stability of an individual's behavioral tendency, other factors including profound life or death experiences (such as bereavement, serious illness, witnessing death-related events), a sense of meaning in life, and family and cultural experiences may all influence death coping ability. Personality traits may arise from both innate and acquired factors [52]. Improved

understanding of nurses' personality traits should therefore benefit future practice, particularly for training and clinical guidance with respect to individual personalities, and for the fostering of acquired personality factors that are well-adapted to the demands of patient care [53].

The self-evaluation score of the nurses' DCS scores was generally at an intermediate level. However, given this was assessed by questionnaire, a self-serving bias is possible wherein subjects increase their scoring to modulate self-esteem [54]. Hospice care showed the highest mean score among the DCSS dimensions. This indicates that nurses pay more attention to understanding and respecting the needs of patients, attach more importance to psychosocial care, and internalize the results of their professional training [11]. Nursing itself is a representative group of the helping industry, with the goal of creating a therapeutic environment with a caring atmosphere. Such an environment helps patients and their families to resolve negative emotional reactions. This is achieved by meeting the needs of patients with characteristics such as respect, focus and care.

The lowest scores on the grief coping dimension of the DCSS were 2.41 for the item, "I can cope with the death of my child", and 2.47 for, "I can cope with the death of my spouse". These were both below the mean of 3.39. It has been reported that, following the death of their loved ones, clinical nurses deal less well with death, are prone to negative emotional reactions, and are less able to cope with it calmly [55]. Furthermore, nurses are always expected to help others, while their own sadness and loss is often ignored [56, 57]. A social support system for nurses that includes colleagues, team members, leaders and family members can alleviate pain in the face of loss through the channels of emotional expression [33]. Intervention nursing managers should note the many ways of strengthening training and promoting grief coping ability. These include discussions of near-death treatment, talking about death, death acceptance, loss treatment, life meaning and the funeral process. Improving the nurses' general ability to cope with expected death and sudden death should therefore improve the quality and efficiency of nursing management.

In the dimension of preparation for death in the DCSS, items such as, "I will buy the contract before I live" and "I will buy the location of my own graveyard or spiritual bone pagoda" scored the lowest. This may be explained by the fact that 93.5% of the participants in this study were ≤40 years old, and that 38.5% of nurses had not received death education. In line with China's cultural background, nurses tend to suppress their true feelings or avoid talking about death issues when facing cadavers, ritual arrangements or pre-preparation projects. The content of educational courses should therefore include the differences in handling death by different religions and cultures.

Donne *et al.* [58] have suggested that including palliative care in nursing education could not only improve professional knowledge and skills, but would also improve attitudes towards palliative care. Therefore, nursing educators should set up palliative care nursing courses and introduce palliative care nursing knowledge into textbooks so that clinical practice planning and theoretical teaching can be achieved systematically [59]. Various teaching methods such as role playing, clinical simulation, case teaching and multimedia teaching can be used to improve the cognitive ability of nursing students in dealing with the dying and death, as well as their ability to communicate with critically ill patients and their families [60, 61]. Hospitals should also encourage and support nurses to learn about palliative care nursing, rigorously carry out the continuing education of in-service nursing staff, and hold regular multi-level, multi-mode, multi-channel palliative care nursing courses and expert lectures. These programs should help to spread palliative care nursing concepts, principles and ethics, pain and symptom control [62]. Because nurses who care for dying patients are psychologically vulnerable to varying degrees, hospitals should offer counseling and psychological knowledge training. This would help nurses to develop effective emotional self-management and to overcome possible

negative mental states [63]. Palliative care nursing medical teams should also be established and the number of palliative care nursing staff increased to ensure the development of this specialty. This would improve not only the quality of life for patients with advanced illness, but also the well-being of nursing staff [64].

## Limitations, strengths and future research

Various limitations of this study should be noted. A low response rate may have limited the generalizability of our findings. In addition, pediatric, neonatal, and obstetric nurses were not included. While the experience of nurses in these departments may affect their ability to cope with death, their inclusion in future research could broaden our understanding of the association between personality and DCS. Convenience sampling was adopted in this study, thus limiting to some extent the representativeness of the sample. Random cluster sampling could be adopted in future studies to reduce selection bias and facilitate better inference of the results. The theoretical basis of this study focussed on self-efficacy. However, the range of variables studied was insufficiently comprehensive given the possible importance of other factors in explaining the variability of coping, such as profound life- or death-related experiences (e.g. bereavement, serious illness, witnessing death-related events), a sense of meaning in life, and family and cultural factors. These factors should be included in larger and more in-depth studies in future.

## Conclusion

Personality traits and DCS are interrelated. In this study, we found that personality traits of agreeableness, openness and conscientiousness were positively correlated with DCS. Hospital managers should pay attention to differences in the personality characteristics of nurses, understand the self-efficacy characteristics of nurses with different personalities, and use this knowledge to carry out targeted professional training. Importantly, however, the assessment of nurses' personality traits should not be used to screen and select nursing candidates. Rather, this information should help them to recognize their reactions in specific situations. Through clinical education, this assessment should also help nurses deal with their emotions in the face of death and adopt better coping methods. This study highlights the importance of preparing nurses to make decisions in the real-world clinical setting with respect to both cognitive and emotional domains.

## Supporting information

**S1 Checklist.**
(DOCX)

**S2 Checklist.**
(DOCX)

**S1 Dataset. The dataset from which the results of the study were produced (SPSS file).**
(SAV)

**S1 File. The data collection tool (questionnaire) in English.**
(DOCX)

## Acknowledgments

The researchers would like to express their gratitude to the nurses who contributed to this study.

## Author Contributions

**Conceptualization:** Qing Liu, Shengwen Shao.

**Data curation:** Xi Lin, Weilan Xiang.

**Formal analysis:** Xiaoqing Li.

**Investigation:** Xi Lin, Weilan Xiang.

**Methodology:** Xiaoqing Li, Qing Liu, Shengwen Shao.

**Project administration:** Shengwen Shao, Weilan Xiang.

**Software:** Xiaoqing Li.

**Supervision:** Qing Liu, Shengwen Shao, Weilan Xiang.

**Writing – original draft:** Xi Lin.

**Writing – review & editing:** Xiaoqing Li, Qing Liu, Shengwen Shao.

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
