## [Decision Letter · Decision Letter 0]

7 Dec 2020

PONE-D-20-32690

A big-five personality model-based study of death coping self-efficacy in clinical nurses

PLOS ONE

Dear Dr. Liu,

Thank you for submitting your manuscript to PLOS ONE. After careful consideration, we feel that it has merit but does not fully meet PLOS ONE’s publication criteria as it currently stands. Therefore, we invite you to submit a revised version of the manuscript that addresses the points raised during the review process.

Whether or not your manuscript can be published will depend, especially, on your response to Reviewer 1's ethical concerns regarding your suggestion that nurses' personality is malleable and should be monitored and shaped by nurse managers. This appears to be a key part of the rationale for your study as well as the most important implication from your results for clinical practice, yet I agree with Reviewer 1 that it is highly problematic from an ethical perspective.

Please also take on board comments from both reviewers that highlight the need to edit your manuscript throughout for English. Unfortunately, PLOS One lacks the resources to edit manuscripts in this way, so responsibility lies with the authors to seek their own support (professional if necessary) in this regard.

We look forward to receiving your revised manuscript.

Kind regards,

Tim Luckett

Academic Editor

PLOS ONE

2. Please clarify in your Methods section whether the questionnaires are published under a CC-BY license, or whether you obtained permission from the publisher to reproduce the questionnaire in this manuscript. Please explain any copyright or restrictions on this questionnaire.

- https://www.sciencedirect.com/science/article/abs/pii/S0020748918300221?via%3Dihub

- https://link.springer.com/article/10.1007/s12282-019-00954-7

- https://onlinelibrary.wiley.com/doi/abs/10.1002/pon.5377

- https://bmchealthservres.biomedcentral.com/articles/10.1186/s12913-018-3478-y

In your revision ensure you cite all your sources (including your own works), and quote or rephrase any duplicated text outside the methods section. Further consideration is dependent on these concerns being addressed.

Reviewers' comments:

Reviewer's Responses to Questions

**Comments to the Author**

1. Is the manuscript technically sound, and do the data support the conclusions?

Reviewer #1: Partly

Reviewer #2: Partly

2. Has the statistical analysis been performed appropriately and rigorously? 

Reviewer #1: N/A

Reviewer #2: No

3. Have the authors made all data underlying the findings in their manuscript fully available?

Reviewer #1: No

Reviewer #2: Yes

4. Is the manuscript presented in an intelligible fashion and written in standard English?

Reviewer #1: No

Reviewer #2: No

5. Review Comments to the Author

Reviewer #1: Dear Editor and Authors

Thanks for the opportunity to read this article. I enclose some comments. Hopefully the researchers will find them useful.

There is no need to include statistics in the abstract. Delete the betas. No need to list tools in the abstract.

The authors claim that they examine how personality profiles have an impact on nursing personnel. At the same time they make use of a regression that does not examine impact but only relationship.

Please write the word China with a capital letter. For some reason there are sentences where the capital letter appears where it should not. The article requires general editing, it has many small shortcomings of punctuation and inaccuracies. The translation of the article in several places seems to have failed in terms of conveying the original meaning to which the researchers aimed. I recommend using a professional language editor.

The authors say: "Moreover, frequent contact between nurses and patients with death-related events can also lead to sympathetic fatigue and exhaustion". There is no such term as sympathetic fatigue. It's compassion fatigue. And this is not exhaustion but burnout. See the article you cited.

" The authors say: Nurses who cannot cope with a patient's death may be unable to support dying patients and their families and minimize the quality of hospice care". Why do you think supporting the family is the role of the nurse? Why not of a social worker, a psychologist? Do the nurses in China undergo professional training to provide emotional support?

*There is no in-depth explanation of personality traits in the article, although this is the main variable

**Study aims should not appear in the method chapter

Study design - not sufficiently described

In the article, the researchers deal with associations of demographic and other factors with death coping self-efficacy of nurses, but in practice a large proportion of these variables were not discussed in the literature review.

The researchers claim that for nursing students and young nurses, their personality is highly malleable, which is precisely the period to establish a correct outlook on life and values and from independent personality. This is a very strange claim. Do you think it is moral and proper to manage or try to change a person's personality in order to adapt it to the needs of the organization? Do you believe this is possible at all?

I feel the researchers included too many variables in the study just because they tested them. It was difficult to find a coherent line of thought in their article. Starting with the presentation of the rationale and its development.

Much has been said in the article about what nurses need to do, but there is little reference to the organization's commitment to nurses. For example institutionalized organizational support.

It is not clear what all the details in Table 2 are for.

Table 4 Factors Affecting death coping self-efficacy (N = 572) - No affect was examined in this study. This is not correct terminology.

In summary the researchers claim that they made use of using hierarchical regression. But in Table 4 this is certainly not the way to present a hierarchical regression. Where are the blocks? There is no need to present Tolerance and VI together. This is a multiplicative inverse.

Reviewer #2: An interesting paper addressed a very important topic but with several major flaws. Below are my comments:

1. Abstract: in the abstract you mentioned that data were collected in June and July 2020, but in the Methods section, you reported it was done in August and September 2020. So, which is right?

2. The section of Introduction is comprehensively organized, but the language needs editing.

3. Methods:

1) Regarding the sample, did you include nurses from pediatric, neonates, and maternity departments? The nurses from these department have different experience and may exhibit a different level of competency on coping with death. You also included ICU nurses, and to my knowledge, these nurses also have different competency of dealing with death. However, I did not understand why you did not collect the data of the departments where nurses work. Why did you include such a different sample but did not analyze the variable of departments, discuss the differences? As you have already finished your data collection, it is impossible to analyze the above questions. Therefore, you may need to mention this as a major limitation in the manuscript.

2) The scale of Death Coping Self-efficacy Scale was translated and revised by a scholar in Taiwan, and it is written in Complex Chinese but not simplified Chinese. More importantly, the people from mainland China where your study was conducted may hold a different perspective in terms of death topics with people in Taiwan. Considering the two facts, I am wondering why you did not translate and test this scale in a sample and report its validity and reliability?

3) Data analysis: in the abstract, you mentioned that you used hierarchical regression analysis, however, in the Methods section, you used multiple regression analysis. From the table, it seemed you used multiple regression analysis. This is contradictory.

4. Results:

What was the mean, SD of score for DCSS? It appears that multiple subgroup analyses based on ANOVA or F tests, t-test or z-test were performed. Given the purpose of the data analyses summarized, I would like to recommend that only multivariate techniques be used such as regression analyses instead of multiple subgroup analyses based on univariate tests in order to assess the contribution of each explanatory variable in estimating the response or outcome variable based on the relationship. Due to the flawed statistical analysis and methods, the article needs to be reworked in a major way and revised before the statistical methods and data analyses can be reviewed and assessed further.

Further to the results, the tables were in a chaos. Re-organize them and present them in a clear manner.

5. Implications to clinical practice and research should be provided.

6. This manuscript is badly written, and there are a number of grammar mistakes. A competent editor is strongly recommended.

6. PLOS authors have the option to publish the peer review history of their article (what does this mean?). If published, this will include your full peer review and any attached files.

Reviewer #1: No

Reviewer #2: No

---

## [Author Response · Author response to Decision Letter 0]

14 Jan 2021

Responds to the academic editor comments:

 We appreciate the academic editor very much for his positive comments and suggestions on our manuscript. At the same time, we are very sorry for the expression is not very clear in this paper.

1. We agree with Reviewer 1, and we have deleted “nurses' personality is malleable and should be monitored and shaped by nurse managers” from the article.

2. We have modified grammatical and syntax errors in our article carefully, with the help of an English translator.

3. The manuscript has been modified according to PLOS ONE's style requirements and file naming requirements. We have made correction which we hope meet with approval.

4. Thank you very much for this helpful suggestion. In this study，Chinese Big Five Personality Inventory brief version has been published publicly in a Chinese journal, and the questionnaire can be reproduced without authorisation from the authors. Death Coping Self-Efficacy Scale，we wrote to the author of the questionnaire, Professor Lizhong Zhang, by E-mail, stating that the use of the scale for academic purpose had been approved by the author. This copyright on this questionnaire has been mentioned in the Methods section. The following picture is the screenshot of the email in which Professor Lizhong Zhang agreed to authorize the use of the questionnaire.

5. We appreciate this suggestion, and we have carefully considered all every change made point by point. We have made correction which we hope meet with approval. Revised portion is marked in blue in the paper.

6. The authors have reviewed the final version of the manuscript and approve it for publication. To the best of our knowledge and belief, this manuscript has not been published in whole or in part nor is it being considered for publication elsewhere.

Responds to the reviewer’s comments:

Response #1:

 I am very grateful to your comments for the manuscript. They are of great importance to us and our article. According to your advice, we amended the relevant part in manuscript. Some of your questions are answered below.

1. It is really true as the reviewer suggested that “there is no need to include statistics in the abstract. Delete the betas. No need to list tools in the abstract”. Therefore, we deleted the statistics, betas, and tools in Abstract.

2. We appreciate this suggestion, and we have used multiple regression analysis to assess the predictors of personality traits on death coping self-efficacy (Table 5).

3. We have modified grammatical and syntax errors in our article carefully, with the help of an English translator.

4. The former statements are inconsiderate. Under your advice, we rewrote the sentence in the text.

5. Thank you very much for this helpful suggestion. In China, nurses, as the implementer, evaluator and educator of hospice care, play an important role in the comprehensive care of end-stage patients and their families. However, due to resource limitations, there is currently a shortage of psychologists and social care workers or volunteers in multidisciplinary teams in hospitals, and very few nurses are professionally trained to provide emotional support. Different emotional experiences of fearfulness, guilt and self-blame appear to emerge in nurses after patient death in a variety of ways, suggesting that nurses lacked the ability to cope with death. We had also discussed these findings in our paper and showed in the Introduction section.

6. We appreciate the reviewer very much for his positive comments and suggestions on our manuscript. At the same time, we are very sorry for the expression is not very clear in this paper. We have added an explanation of personality in the Introduction and Discussion section.

7. Under your advice, we have deleted the research purpose in the method chapter. 

8. Under your advice, we have supplemented study design in the Methods section.

9. We appreciate this suggestion. We have added the relationship between demographic and other factors and death coping self-efficacy of nurses in the Introduction section.

10. We appreciate the reviewer very much for his positive comments and suggestions on our manuscript. At the same time, we are very sorry for the expression is not very clear in this paper. We agree with Reviewer 1, and we have deleted “nurses' personality is malleable and should be monitored and shaped by nurse managers” from the article.

11. Under your advice, we have removed demographic variables from multiple regression analysis, and we have re-written the "introduction" with the rationale and development of death coping self-efficacy and personality traits.

12. This is a very valuable recommendation, we didn’t think about it before. Therefore, we added institutionalized organizational support in the Discussion section.

13. Under your advice, Table 2 has been revised to death coping self-efficacy mean score of nurses.

14. We appreciate this suggestion, and we have modified it to be predictors of nurses’ death coping self-efficacy (Table 5).

15. Thank you very much for this helpful suggestion. We have modified the statistical method to multiple regression analysis (Table 5), and we have deleted VI in Note. Revised portion are marked in blue in the paper.

Response #2:

We also appreciate the reviewer very much for his positive comments and suggestions on our manuscript. 

1. Abstract:

 We are very sorry for the expression is not very clear in this paper, and we have amended it in the abstract and the data collection was completed in August and September 2020.

2. We have modified grammatical and syntax errors in our article carefully, with the help of an English translator.

3. Methods:

1）This is a very valuable recommendation, we didn’t think about it before. We have explained a discussion of the limitations of the non-inclusion of pediatric, neonates, and maternity departments.

2) We appreciate the reviewer very much for his positive comments and suggestions on our manuscript. At the same time, we are very sorry for the expression is not very clear in this paper. Before the questionnaire survey, cross-cultural adjustment, pre-survey and validity and reliability test of the scale were carried out, which were not included in the paper before. Sample translations and tests of the scale, and reports on its validity and reliability have been supplemented in the method.

3） Under your advice, we have modified it for multiple regression analysis in the Methods section.

4. Results:

It is really true as reviewer suggested that only multivariate techniques be used such as regression analyses instead of multiple subgroup analyses based on univariate tests in order to assess the contribution of each explanatory variable in estimating the response or outcome variable based on the relationship. Therefore, we have used multiple regression analysis to assess the predictors of personality traits on death coping self-efficacy (Table 5). Demographic variables have been removed from the Table 5 and modified in the corresponding places in the article. 

5.We have carefully considered all every change made point by point. We have made corrections which we hope meet with approval. Revised portion are marked in blue in the paper.

We tried our best to improve the manuscript and made some changes in the manuscript. These changes will not influence the content and framework of the paper. And here we did not list the changes but marked in blue in revised paper. We appreciate for Reviewers’ warm work earnestly, and hope that the correction will meet with approval.

 Once again, thank you very much for your comments and suggestions.

Looking forward to hearing from you. 

Thank you and best regards.

Yours sincerely,

 Xi Lin

---

## [Decision Letter · Decision Letter 1]

10 Feb 2021

PONE-D-20-32690R1

Big Five personality model-based study of death coping self-efficacy in clinical nurses: A cross-sectional survey

PLOS ONE

Dear Dr. Liu,

Thank you for submitting your manuscript to PLOS ONE. After careful consideration, we feel that it has merit but does not fully meet PLOS ONE’s publication criteria as it currently stands. Therefore, we invite you to submit a revised version of the manuscript that addresses the points raised during the review process.

We look forward to receiving your revised manuscript.

Kind regards,

Tim Luckett

Academic Editor

PLOS ONE

Additional Editor Comments:

General comments

It is important that in your rebuttal letter, you include the reviewer’s comments as well as your attempt to address them, as in your last letter it was difficult to work out which of the reviewer’s comments each of your responses referred to.

While the English grammar has been substantially improved in the revised version, the changes have not always been effectively instituted, with some remnants of previous wording inappropriately left in alongside the revised text. I suspect this is simply because the track changes can make it difficult to proof read the new sentence; please start by accepting the track changes to date and then add new track changes as appropriate.

While I’m pleased to see the authors have changed problematic wording such as “manipulated”, I am still concerned about their rationale that investigating the relationship between personality and self-efficacy in death coping can inform “selecting nurses with appropriate personality traits to join the hospice care team”. This invests far too much confidence in the results of a cross-sectional study that can only provide insight into correlations between a limited range of variable, as well as the validity of the Big 5 Personality Test for measuring personality characteristics most relevant to EOL nursing. After all, the model only explained 20% of the variation in coping, as highlighted in the Discussion. Please remove this idea altogether from both the Introduction, Discussion and Conclusion, and focus on the second part of the rationale regarding education and counselling. The Conclusion in particular will effectively need to be rewritten.

There are still instances throughout where the language inappropriately suggests causal relationships that cannot be inferred from cross-sectional data. These include “determined”, “influenced” and “affected”. Strictly speaking, “predict” should also not be used for cross-sectional data since it implies a future time.

Please use “Big Five Personality Test” or “Model” consistently throughout rather than varying capitalisation and the final word.

Introduction

Please break up the very long paragraph that begins with “Bandura”.

The sentence beginning “In contrast” is difficult to follow.

“We question whether” is an English phrase usually used to express disbelief, when I think the authors really mean “it is unknown whether”?

In the sentence beginning “the sense of responsibility” and the following one, please clarify whether these studies were undertaken with nurses.

Discussion

Please rephrase the sentence beginning “For nurses faced with patients” to improve grammar.

The sentence beginning “The Big Five personality has a significant predictive” needs substantial rewriting to clarify meaning and avoid appearing to be contradictory. Please also remove “poor” as a description of correlation.

Please replace “universality” with “generalizability”.

Please add to the limitations section with further acknowledgement that only a limited range of variables were measured, and that some others (as highlighted earlier in the Discussion) might also be important in explaining variability.

Reviewers' comments:

Reviewer's Responses to Questions

**Comments to the Author**

1. If the authors have adequately addressed your comments raised in a previous round of review and you feel that this manuscript is now acceptable for publication, you may indicate that here to bypass the “Comments to the Author” section, enter your conflict of interest statement in the “Confidential to Editor” section, and submit your "Accept" recommendation.

Reviewer #2: All comments have been addressed

2. Is the manuscript technically sound, and do the data support the conclusions?

Reviewer #2: Partly

3. Has the statistical analysis been performed appropriately and rigorously? 

Reviewer #2: Yes

4. Have the authors made all data underlying the findings in their manuscript fully available?

Reviewer #2: Yes

5. Is the manuscript presented in an intelligible fashion and written in standard English?

Reviewer #2: No

6. Review Comments to the Author

Reviewer #2: Thank you for revising this manuscript. A few more questions arise.

(1) on page 2, Introduction, ‘Robbins extracted four dimensions of death coping self-efficacy…’ please provide the reference.

(2) In the Introduction, you used ‘hospice care’ and ‘palliative care’. These two are different, and you were supposed to focus on just one term, so you need to delete or change the other. I think you were focusing on palliative care setting. Too much redundant information was provided in Introduction, but little was about personality trait and its relationship with death coping self-efficacy. My suggestion is to shorten this section and focus on what you were doing.

(3) You were supposed to mark the changes in red, to enable the reviewers easily understand what you revised in the manuscript. I am confused about the documents you provided for further revision, as I am not sure which document is the final one.

(4) There is a big flaw---in the Abstract, you mentioned that you focused on palliative care settings, however, in the Methods, you recruited nurses from general wards and ICU. These are not palliative care setting.

(5) A competent English editor is recommended.

7. PLOS authors have the option to publish the peer review history of their article (what does this mean?). If published, this will include your full peer review and any attached files.

Reviewer #2: No

---

## [Author Response · Author response to Decision Letter 1]

11 Mar 2021

Dear Reviewers,

I am very grateful to your comments for the manuscript. They are of great importance to us and our article. According to your advice, we amended the relevant part in manuscript. Revised portion are marked in red in the paper. The main corrections in the paper and the responds to the reviewer’s comments are as flowing:

Responds to the academic editor comments:

Editor:

1. It is important that in your rebuttal letter, you include the reviewer’s comments as well as your attempt to address them, as in your last letter it was difficult to work out which of the reviewer’s comments each of your responses referred to.

Response: We appreciate the reviewer very much for his positive comments and suggestions on our manuscript. At the same time, we are very sorry for the expression is not very clear in this paper. We have adopted a question-and-answer approach to facilitate the review of editors and reviewers.

2. While I’m pleased to see the authors have changed problematic wording such as “manipulated”, I am still concerned about their rationale that investigating the relationship between personality and self-efficacy in death coping can inform “selecting nurses with appropriate personality traits to join the hospice care team”. This invests far too much confidence in the results of a cross-sectional study that can only provide insight into correlations between a limited range of variable, as well as the validity of the Big 5 Personality Test for measuring personality characteristics most relevant to EOL nursing. After all, the model only explained 20% of the variation in coping, as highlighted in the Discussion. Please remove this idea altogether from both the Introduction, Discussion and Conclusion, and focus on the second part of the rationale regarding education and counselling. The Conclusion in particular will effectively need to be rewritten.

Response: Thank you very much for this helpful suggestion. We have removed the idea in Introduction, Discussion and Conclusion, and have focused on the second part of the rationale regarding education and counselling. ( line 405-425, pages 19 and 20). 

The conclusion has been rewritten.

3.There are still instances throughout where the language inappropriately suggests causal relationships that cannot be inferred from cross-sectional data. These include “determined”, “influenced” and “affected”. Strictly speaking, “predict” should also not be used for cross-sectional data since it implies a future time.

Response: Thank you very much for this helpful suggestion, and we have removed the word "predict" from the article.

4. Please use “Big Five Personality Test” or “Model” consistently throughout rather than varying capitalisation and the final word.

Response: Under your advice, we revised the corresponding place in the paper

5. Introduction

Please break up the very long paragraph that begins with “Bandura”.

Response: Under your advice, we broken down a long passage at the beginning of "Bandura.

6.The sentence beginning “In contrast” is difficult to follow.

Response: We have modified the sentence. ( line 102-104, pages 5) 

7. “We question whether” is an English phrase usually used to express disbelief, when I think the authors really mean “it is unknown whether”?

Response: Under your advice, we modified it to "it is unknown whether". ( line 144, pages 6) 

8. In the sentence beginning “the sense of responsibility” and the following one, please clarify whether these studies were undertaken with nurses.

Response: Under your advice, we clarified that these studies were conducted by nurses. ( line 115-118, pages 5) 

9. Discussion

Please rephrase the sentence beginning “For nurses faced with patients” to improve grammar.

Response: Under your advice, we rewrote the sentence beginning “For nurses faced with patients”. ( line 334-335, pages 16)

10.The sentence beginning “The Big Five personality has a significant predictive” needs substantial rewriting to clarify meaning and avoid appearing to be contradictory. Please also remove “poor” as a description of correlation.

Response: Under your advice, we rewrote the sentence that begins with " The Big Five personality has a significant predictive " and changed the word "poor" to "weak." ( line 354-361, pages 17)

11. Please replace “universality” with “generalizability”.

Response: Under your advice, we have replaced "universality" with "generalizability". ( line 429, pages 20)

12. Please add to the limitations section with further acknowledgement that only a limited range of variables were measured, and that some others (as highlighted earlier in the Discussion) might also be important in explaining variability.

Response: Under your advice, we added this note in the limitations section. ( line 435-441, pages 20 and 21)

Responds to the reviewer’s comments:

Reviewer #2:

We also appreciate the reviewer very much for his positive comments and suggestions on our manuscript. 

1. On page 2, Introduction, ‘Robbins extracted four dimensions of death coping self-efficacy…’ please provide the reference.

Response: Under your advice, we provided references in the introduction ‘Robbins extracted four dimensions of death coping self-efficacy…’. ( line 72-74, pages 4)

2. In the Introduction, you used ‘hospice care’ and ‘palliative care’. These two are different, and you were supposed to focus on just one term, so you need to delete or change the other. I think you were focusing on palliative care setting. Too much redundant information was provided in Introduction, but little was about personality trait and its relationship with death coping self-efficacy. My suggestion is to shorten this section and focus on what you were doing.

Response: This is a very valuable recommendation, we didn’t think about it before. We removed the 'hospice care', and added information about personality trait and its relationship with death coping self-efficacy. ( line 124-140, pages 6)

3. You were supposed to mark the changes in red, to enable the reviewers easily understand what you revised in the manuscript. I am confused about the documents you provided for further revision, as I am not sure which document is the final one.

Response: We are very sorry for the expression is not very clear in this paper. Revised portion are marked in red in the paper.

4. There is a big flaw---in the Abstract, you mentioned that you focused on palliative care settings, however, in the Methods, you recruited nurses from general wards and ICU. These are not palliative care setting.

Response: We appreciate this suggestion. Because of hospice care in China starts late, only in 2019 opened the first hospice care specialist nurse training, so few hospice care specialist nurse, palliative care environment is not mature, so can only be recruited from the common ward and the ICU nurses, for hospital and community hospice care team, choose a suitable personality traits nurses into the hospice care team, improve the level of hospice care.

5. A competent English editor is recommended. 

Response: We have carefully considered all every change made point by point. We have made correction which we hope meet with approval.

---

## [Decision Letter · Decision Letter 2]

12 Apr 2021

PONE-D-20-32690R2

Big Five Personality Model-based study of death coping self-efficacy in clinical nurses: A cross-sectional survey

PLOS ONE

Dear Dr. Liu,

Thank you for submitting your manuscript to PLOS ONE. After careful consideration, we feel that it has merit but does not fully meet PLOS ONE’s publication criteria as it currently stands. Therefore, we invite you to submit a revised version of the manuscript that addresses the points raised during the review process.

We look forward to receiving your revised manuscript.

Kind regards,

Tim Luckett

Academic Editor

PLOS ONE

Journal Requirements:

Additional Editor Comments (if provided):

I am pleased to see the authors have reduced the inference that personality testing should be used to assess nurses’ suitability for PC. However, there remains an instance of this in the aim (“to provide a basis for selecting nurses with appropriate personality traits to join the palliative care team”) conclusion (“However, palliative care positions should be assigned according to candidates’ personality traits …”) that also need removing. I also suggest removing “to develop appropriate counselling and educational measures to improve the level of palliative care” from the aim and leaving this implication of the research to the Discussion.

Similarly, the authors have removed instances of ‘predict’ as requested but not all instances of ‘influence’, most notably from the abstract. This should be replaced by ‘associations between’ to less imply a causal relationship.

I was interested in the authors’ response to Reviewer #2’s query regarding whether nurses were specialist PC or not. It needs to be made much clearer that there were no SPC services in China until very recently and explained that this is why the study focused on ward/ICU nurses. Some more background and clarification should be inserted around a revised version of the sentence “In China, nurses, as the implementers, evaluators and educators of palliative care, play an important role in the comprehensive care of end-stage patients and their families”. Then, the aim and Methods need to make clear the focus was on ward/ICU. So, when combined with the suggestions regarding the aim above, the aim should simply read “This study aimed to explore the relationship between personality traits and death coping self-efficacy in nurses from general wards and ICU”.

More specific suggestions include the following:

• Please insert headings into the Abstract – Background, Methods, Results, Conclusions

• Although the authors have removed some instances of ‘hospice’, there remain some instances in the discussion that need changing to ‘palliative care’.

• Please remove ‘questionnaire’ from ‘questionnaire survey design’.

• Please insert an ‘r’ in the word ‘researches’.

I agree with Reviewer 2 that, while the standard of English is generally good, phrasing is frequently a little awkward and would benefit from editing throughout by someone for whom English is a first language.

I would also welcome attempts to shorten the manuscript, which is very long.

Reviewer's Responses to Questions

**Comments to the Author**

1. If the authors have adequately addressed your comments raised in a previous round of review and you feel that this manuscript is now acceptable for publication, you may indicate that here to bypass the “Comments to the Author” section, enter your conflict of interest statement in the “Confidential to Editor” section, and submit your "Accept" recommendation.

Reviewer #2: All comments have been addressed

2. Is the manuscript technically sound, and do the data support the conclusions?

Reviewer #2: Yes

3. Has the statistical analysis been performed appropriately and rigorously? 

Reviewer #2: Yes

4. Have the authors made all data underlying the findings in their manuscript fully available?

Reviewer #2: Yes

5. Is the manuscript presented in an intelligible fashion and written in standard English?

Reviewer #2: No

6. Review Comments to the Author

Reviewer #2: Thank you for your revision. I think the authors have addressed all the issues raised by the reviewers.

One question I think at this moment is to have this manuscript checked by an English speaker for the language, to make it more clear, and unambiguous.

7. PLOS authors have the option to publish the peer review history of their article (what does this mean?). If published, this will include your full peer review and any attached files.

Reviewer #2: No

---

## [Author Response · Author response to Decision Letter 2]

6 May 2021

Dear Reviewers,

I am very grateful to your comments for the manuscript. They are of great importance to us and our article. According to your advice, we amended the relevant part in manuscript. Revised portion are marked in red in the paper. The main corrections in the paper and the responds to the reviewer’s comments are as flowing:

Responds to the academic editor comments:

Editor:

Response: We appreciate the reviewer very much for his positive comments and suggestions on our manuscript. We have reviewed the reference list to ensure that it is complete and correct.

2. Additional Editor Comments (if provided):

I am pleased to see the authors have reduced the inference that personality testing should be used to assess nurses’ suitability for PC. However, there remains an instance of this in the aim (“to provide a basis for selecting nurses with appropriate personality traits to join the palliative care team”) conclusion (“However, palliative care positions should be assigned according to candidates’ personality traits …”) that also need removing. I also suggest removing “to develop appropriate counselling and educational measures to improve the level of palliative care” from the aim and leaving this implication of the research to the Discussion.

Response: Thank you very much for this helpful suggestion. We have removed the inference in the purposes and conclusions that personality testing should be used to assess nurses’ suitability for PC.

We have removed “to develop appropriate counselling and educational measures to improve the level of palliative care” from the aim and leaving this implication of the research to the Discussion. ( line 328-330, pages 18)

3. Similarly, the authors have removed instances of ‘predict’ as requested but not all instances of ‘influence’, most notably from the abstract. This should be replaced by ‘associations between’ to less imply a causal relationship.

Response: Thank you very much for this helpful suggestion, and we have replaced 'influence' with 'associations between'. ( line 28, pages 2) 

4. I was interested in the authors’ response to Reviewer #2’s query regarding whether nurses were specialist PC or not. It needs to be made much clearer that there were no SPC services in China until very recently and explained that this is why the study focused on ward/ICU nurses. Some more background and clarification should be inserted around a revised version of the sentence “In China, nurses, as the implementers, evaluators and educators of palliative care, play an important role in the comprehensive care of end-stage patients and their families”. 

Then, the aim and Methods need to make clear the focus was on ward/ICU. So, when combined with the suggestions regarding the aim above, the aim should simply read “This study aimed to explore the relationship between personality traits and death coping self-efficacy in nurses from general wards and ICU”.

Response: We are very sorry for the expression is not very clear in this paper. Some more background and clarification have been inserted around a revised version of the sentence “In China, nurses, as the implementers, evaluators and educators of palliative care, play an important role in the comprehensive care of end-stage patients and their families”. ( line 79-90, pages 4 and 5)

This is a very valuable recommendation, we didn’t think about it before.We have made it clear in the aim and methods that the focus is on the ward /ICU. ( line 158, pages 8)

5. Please insert headings into the Abstract – Background, Methods, Results, Conclusions.

Response: Under your advice, we have inserted headings into the Abstract – Background, Methods, Results, Conclusions. ( line 24-43, pages 2)

6. Although the authors have removed some instances of ‘hospice’, there remain some instances in the discussion that need changing to ‘palliative care’.

Response: We appreciate this suggestion. We have changed the word ‘hospice’ in the discussion to ‘palliative care’. ( line 438-447, pages 23)

7. Please remove ‘questionnaire’ from ‘questionnaire survey design’.

Response: Under your advice, we have removed the word ‘questionnaire’ from the ‘questionnaire survey design’.( line 179, pages 9) 

8. Please insert an ‘r’ in the word ‘researches’.

Response: Under your advice, we have used the word ‘studies’ instead of ‘researches’. ( line 150, pages 7) 

9. I agree with Reviewer 2 that, while the standard of English is generally good, phrasing is frequently a little awkward and would benefit from editing throughout by someone for whom English is a first language.

Response: We apologize for the poor language of our manuscript. We worked on the manuscript for a long time and the repeated addition and removal of sentences and sections obviously led to poor readability. We have now worked on both language and readability and have also involved native English speakers for language corrections. We really hope that the flow and language level have been substantially improved.

10. I would also welcome attempts to shorten the manuscript, which is very long.

Response: We have tried to shorten the text wherever possible, also in line with the editors' comments. Editors are also welcome to shorten manuscripts.

---

## [Editor Report · Decision Letter 3]

11 May 2021

PONE-D-20-32690R3

Big Five Personality Model-based study of death coping self-efficacy in clinical nurses: A cross-sectional survey

PLOS ONE

Dear Dr. Liu,

Thank you for submitting your manuscript to PLOS ONE. The authors have done  good job of addressing previous comments. However, please remove the following sentence from the Discussion which is misleading about the aim of the study (and which is not normally restated in a Discussion anyway): "The aim was to develop appropriate counseling and educational measures in order to improve the quality of palliative care".

A rebuttal letter that responds to the point raised above. You should upload this letter as a separate file labeled 'Response to Reviewers'.A marked-up copy of your manuscript that highlights changes made to the original version. You should upload this as a separate file labeled 'Revised Manuscript with Track Changes'.An unmarked version of your revised paper without tracked changes. You should upload this as a separate file labeled 'Manuscript'.

We look forward to receiving your revised manuscript.

Kind regards,

Tim Luckett

Academic Editor

PLOS ONE

---

## [Author Response · Author response to Decision Letter 3]

11 May 2021

Dear Reviewers,

I am very grateful to your comments for the manuscript. They are of great importance to us and our article. According to your advice, we amended the relevant part in manuscript. Revised portion are marked in red in the paper. The main corrections in the paper and the responds to the reviewer’s comments are as flowing:

Responds to the academic editor comments:

Editor:

1. Please remove the following sentence from the Discussion which is misleading about the aim of the study (and which is not normally restated in a Discussion anyway): "The aim was to develop appropriate counseling and educational measures in order to improve the quality of palliative care".

Response: We appreciate the reviewer very much for his positive comments and suggestions on our manuscript. We have removed the following sentence from the Discussion: "The aim was to develop appropriate counseling and educational measures in order to improve the quality of palliative care".

---

## [Editor Report · Decision Letter 4]

17 May 2021

Big Five Personality Model-based study of death coping self-efficacy in clinical nurses: A cross-sectional survey

PONE-D-20-32690R4

Dear Dr. Liu,

We’re pleased to inform you that your manuscript has been judged scientifically suitable for publication and will be formally accepted for publication once it meets all outstanding technical requirements.

Kind regards,

Tim Luckett

Academic Editor

PLOS ONE

---

## [Editor Report · Acceptance letter]

19 May 2021

PONE-D-20-32690R4 

Big Five Personality Model-based study of death coping self-efficacy in clinical nurses: A cross-sectional survey 

Dear Dr. Liu:

I'm pleased to inform you that your manuscript has been deemed suitable for publication in PLOS ONE. Congratulations! Your manuscript is now with our production department. 

Kind regards, 

on behalf of

Dr. Tim Luckett 

Academic Editor

PLOS ONE